# A Performance Comparison of Different Cloud-Based Natural Language Understanding Services for an Italian e-Learning Platform [†]

**Matteo Zubani [1], Luca Sigalini [2], Ivan Serina [1,*], Luca Putelli [1,*], Alfonso E. Gerevini [1] and Mattia Chiari [1]**

1    Department of Information Engineering, University of Brescia, Via Branze 38, 25121 Brescia, Italy; m.zubani004@unibs.it (M.Z.); alfonso.gerevini@unibs.it (A.E.G.); m.chiari017@unibs.it (M.C.)
2    Mega Italia Media, Via Roncadelle 70A, 25030 Castel Mella, Italy; luca.sigalini@megaitaliamedia.it
*    Correspondence: ivan.serina@unibs.it (I.S.); l.putelli002@unibs.it (L.P.)
†    This paper is an extended version of our paper "Evaluating Different Natural Language Understanding Services in a Real Business Case for the Italian Language", Published in the 24th International Conference on Knowledge-Based and Intelligent Information & Engineering, Virtual Event, 16–18 September 2020.

**Abstract:** During the COVID-19 pandemic, the corporate online training sector has increased exponentially and online course providers had to implement innovative solutions to be more efficient and provide a satisfactory service. This paper considers a real case study in implementing a chatbot, which answers frequently asked questions from learners on an Italian e-learning platform that provides workplace safety courses to several business customers. Having to respond quickly to the increase in the courses activated, the company decided to develop a chatbot using a cloud-based service currently available on the market. These services are based on Natural Language Understanding (NLU) engines, which deal with identifying information such as entities and intentions from the sentences provided as input. To integrate a chatbot in an e-learning platform, we studied the performance of the intent recognition task of the major NLU platforms available on the market with an in-depth comparison, using an Italian dataset provided by the owner of the e-learning platform. We focused on intent recognition, carried out several experiments and evaluated performance in terms of F-score, error rate, response time, and robustness of all the services selected. The chatbot is currently in production, therefore we present a description of the system implemented and its results on the original users' requests.

**Keywords:** chatbots; natural language understanding; cloud-based services; machine learning

## 1. Introduction

In December 2019, COVID-19 spread rapidly around the world and led the governments of different countries to impose limitations on people's lives. The World Health Organization (WHO) declared the coronavirus epidemic a pandemic (World Health Organization, Coronavirus (COVID-19) pandemic. https://www.who.int/emergencies/diseases/novel-coronavirus-2019, accessed on 17 February 2022). Due to the emergency conditions, according to UNESCO Institute for Statistics (UIS) (UNESCO Institute for Statistics data. COVID-19 Impact on Education. https://en.unesco.org/covid19/educationresponse http://data.uis.unesco.org, accessed on 17 February 2022), on 1 April 2020, more than 170 countries decided to close schools and higher education institutions, affecting 84% of the students enrolled, which is little less than 1.5 billion pupils. On 1 December 2020, schools were still closed in 29 countries, which means more than 300 million students (18.2%) could not take face-to-face lessons.

Given the above, it was necessary to quickly find solutions that would allow institutions to continue providing lessons to students through electronic devices. Universities, colleges, and schools resorted to online applications to communicate with students through

virtual environments using webcams and microphones in order to make lessons available electronically. It is evident that e-learning tools and platforms were crucial to continuing to give lessons in the education field.

Although the most common idea of e-learning is related to the educational environment, e-learning can affect many sectors. Several companies offer specific online courses aimed at helping employees to improve their skills or satisfy legal obligations regarding attending workplace safety courses.

The Italian company Mega Italia Media acting in the e-learning sector provides "workplace safety" courses to Italian companies. They gave us interesting data about the courses activated in 2020, and we compared them in Figure 1 with the same data collected in 2018 and 2019. In Figure 1, we indicate with the vertical green dashed lines the beginning and the end of the first Italian lockdown, while we show the introduction of new restrictions (on individuals, schools, and companies) in October using the yellow dotted line. As we can see in the first two months of 2020, the number of courses activated is similar to 2019. During the national lockdown in March and April, the courses activated are significantly more numerous if compared with the same period of the previous years with a peak of +61% in April. The national lockdown pushed companies that were already *Mega Italia Media* customers to increase the online course purchase for their employees who could only work from home, triggering a digital transformation process for business training in many Italian companies. Starting from September and with the new restrictions, the increase was significant and reached a peak in November with +91% of the courses activated in 2019. The massive growth in the last four months of 2020 came from the digital transformation process started in March; in fact, many of the new customers reached during the first lockdown found e-learning useful and convenient, preferring it to the traditional model.

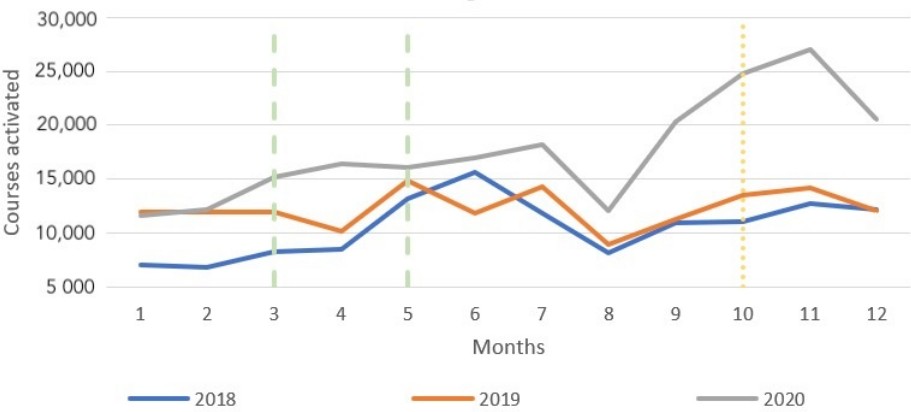

**Figure 1.** The number of courses activated in the years 2018, 2019, and 2020. The vertical lines of green dashes represent the beginning and the end of the first Italian lockdown. The yellow dotted line indicates the introduction of new restrictions in Autumn 2020.

Given the growth in the e-learning sector, the company *Mega Italia Media* decided to introduce innovative solutions based on artificial intelligence on its platform. The company developed a chatbot, a virtual assistant integrated into the chat already present in their platform. There are many definitions for what a *chatbot* (or *chatterbot*, in its original form) is, starting from the simplest one provided by Michael Mauldin in 1994 that defines a chatbot as a computer program that can talk with humans [1]. A more complete and specific definition is provided in [2,3], which defines a chatbot as "an intelligent conversational system that is able to process the human language and formulate a response which has to be relevant to the user input sentence". Although the original idea of a chatbot comes from the Turing test, in which a computer program talks with humans trying to convince them that their interlocutor is not artificial [4], in current applications, chatbots are simply designed to provide information and satisfy humans' requests [5,6], providing them with a more user-friendly interface in comparison, for instance, to a static content search of a

website [7]. In our context, the chatbot has to answer the students' most frequently asked questions, so as to significantly reduce the workload of tutors, who are the ones that usually assist the learners. The chatbot implementation improves the flexibility and scalability of the assistance system implemented, allowing dealings with peaks in requests with few tutors. As reported by [8], from an architectural point of view chatbots are composed by three main modules:

- The **User Interface**, which allows the user to dialogue with the service, by talking or typing their questions.
- The **Application Core**, which contains the logic that allows the dialogue with the user, analyzing the text or the speech, defining the conversation flow and the exchange of information. In this module, the chatbot must keep track of the context and the previous interaction with the user: for example, in the question "How can I activate it?" the chatbot has to reference the pronoun *it* to a previous entity the user has mentioned. This task is often referred as Dialogue State Tracking [9].
- The interface with the **External Services**, which allows the application core to connect with databases, human operators and other services in order to satisfy the users' requests.

Focusing on the structure and implementation of the Application Core, the work in [6] presents a division of chatbots into several categories. The *template-based* chatbots select responses, computing the similarity between the user request and a predefined template using pattern matching techniques. The *corpus-based* chatbots integrate knowledge engineering techniques to store information which can be used to satisfy the users' requests [10] or exploit word embedding techniques [11] for better computing the similarity between questions and answers [12]. Sequence-to-sequence (Seq2Seq) models, in which a deep learning model (such as a Recurrent Neural Network) elaborates the request and then automatically generates the answer, are defined by [6] as *RNN-based* chatbots. However, in our opinion, other Seq2Seq models based on the Transformer architecture [13] (such as Meena by Google [14]) can also be included in this category for their ability to generate an answer without a set of predefined questions. A similar capability but with a different implementation defines the *RL-based* chatbots, which use Reinforcement Learning for response generation.

Given the urgency to implement this kind of system, the company decided to use one of the cloud-based services that allows the fast development of a virtual assistant. Most of these services can be defined (according to the schema presented in [6]) as *intent-based* chatbots. The main activity that an intent-based chatbot has to perform is to recognize the *intentions* of the user, which can be expressed in many different ways. However, other important information can be found in the user request, such as times, addresses and other type of *entities*. Using intentions and entities, a NLU engine can formulate a specific answer that can effectively help the user with their request. Several IT companies have developed cloud-based platforms, offering systems that can be exploited for implementing virtual assistants very rapidly and with no specific knowledge of Natural Language Understanding. Moreover, these platforms can be used with only a small amount of examples, with no need for providing a large quantity of training data.

The goal of this work is to evaluate the performance of the main cloud-based NLU platforms in a real-world context, using the requests collected through the chat service of *Mega Italia Media*. Except for the anonymization and the removal of personal and sensitive data, the utterances were not pre-processed. Once the service was chosen, we used it to develop the new assistance system, and we compared its performance with the benchmark obtained from the previous evaluations. Several works perform this kind of comparison, mainly from an architectural and implementation perspectives and simply showing the results of the intent recognition task [15–17], often considering an already existing application [18–20]; in this paper, we tackle several fundamental aspects that have to be considered in developing a chatbot using a cloud-based platform, such as the number of the training instances for each intention and its relation with the performance. As pointed out in [6], service-oriented chatbots (such as the one we implemented for *Mega*

*Italia Media*) have their specific lexicon and require a customization process which can be based on a very low number of training instances. Moreover, since in terms of usability the response time has to be considered, we compared the response time of the different platforms in order to identify if there are relevant differences between the cloud-based platforms; in addition, we evaluate the performance also in terms of misspellings and errors. Another important aspect of our work is that it is based on the Italian language, which has a limited diffusion. Given that most studies are conducted with a dataset in English, it can be useful to study in the case that there are major drawbacks in the use of chatbots for less common languages.

The structure of this paper is as follows: in Section 2, we overview related works; in Section 3, we describe the considered cloud-based platforms; in Section 4, we describe our case study and the datasets we used for our evaluation; in Section 5, we present the main components of our analysis; in Section 6, we show our results; Section 7 shows the system architecture implemented, and we present the performance in terms of error rate; finally, in Section 8, we put forward our conclusions and future work.

## 2. Related Work

As for many other Natural Language Processing tasks, in the last few years, Natural Language Understanding has made significant progress. Algorithms such as Word2Vec [11] that exploit neural networks for word representation, or the use of Recurrent Neural Networks for processing sentences and documents in order to understand their meaning, have radically improved the results of text classification tasks [21], machine translation [22], or question answering [23]. Moreover, the attention mechanism [24] allows recurrent-based models to find the most relevant words in a document and provides useful insights of the reasoning behind the neural network model [25,26]. Even more recently, attention-based architectures such as Transformer [13] and BERT [27] set a new state-of-the-art for Natural Language Understanding tasks [28–30].

NLU algorithms are used in sensitive and critical environments such as medicine [26,31–36] or in less critical but challenging tasks such as sentiment analysis [37] or the realization of conversational agents [38,39]. For less critical and general-purpose tasks, many commercial NLU platforms (such as chatbots or Spoken Dialogue Systems) have been released in the last few years.

While most of the studies regard sentences and documents that are written in the English language [6], there are important works also focusing on Italian [40]. For instance, in [41] a conversational agent for the Italian public administration is presented. A BERT model for Italian understanding has been presented in [42] and has been applied for hate speech detection [43].

There are several works that show how cloud-based platforms can be used to create virtual assistants in different domains. For instance, ref. [20] implements a medical dialogue system using DialogFlow. In the education domain, a virtual teacher based on Watson is realized by [44]. The work in [45] focuses on the integration of chatbots with the Internet of Things. For the Italian language, in [18] Leoni et al. presented a system based on IBM Watson that provides automatic answers to questions regarding the artificial intelligence domain. However, in these papers, there is no information about how they chose a specific NLU service instead of another one, and the authors do not focus on the analysis of the performance obtained.

In the last few years, comparisons have been made of different datasets by [46], which builds and analyzes two different corpora and in particular by [17], which presents a detailed description and an in-depth evaluation (using a dataset made by utterances concerning the weather) of the most common platforms. Another important analysis was made by [47], which proposed a comparison regarding the performance of four NLU services on very large datasets (with more than 20 domains and 50 intentions). With respect to our analysis, however, their focus is mostly on implementation details (such as the presence of SDK, programming languages, etc.) and they simply report performance in

intent or entity recognition, while we focus also on other important aspects such as response time and the relation with training set instances. Moreover, all of these evaluations are made with datasets composed of text written in English. Given its worldwide diffusion, the models for the English language are the most studied and optimized, while analysis for other languages are definitely rarer. We think that an analysis for a less common language such as Italian can be useful also for other applications in languages such as German, Polish, or Spanish.

Regarding the domain concerning this paper, the work in [48] presents a preliminary analysis of the performance of cloud-based NLU platforms. In this work, we substantially extend that analysis, introducing new results and several improvements, in particular focusing on evaluating the most important characteristics required for an implementation in the real world.

### 3. Natural Language Understanding Cloud Platforms

The main goal of NLU services is the extraction of useful and structured information from an unstructured input like a natural language sentence or document. In general, NLU services focus on two aspects:

- **Intent**—the services have the goal to understand what the user means in their sentence, which can be any type of utterance (greetings, requests, complaints, etc.), and provide a classification value for it.
- **Entity**—the services should identify the important parameters of the users' requests, extracting names, addresses, numbers, quantities, etc.

To clarify these two aspects, Figure 2 shows the output of IBM Watson using the sentence "Do you want to go out tomorrow?" as an input example. In Figure 2, labeled with number 1, the platform classifies the intention as "hang_out", which is the meaning of the user request. In label 2 of Figure 2, the NLU engine extracts the word "tomorrow" and labels it as an entity with type "date". Please note that the platform automatically converts the word "tomorrow" to a specific date in a structured format.

The most used NLU platforms on the market are:

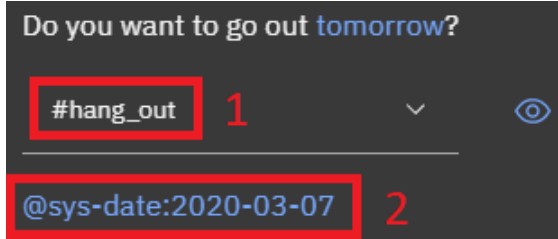

**Figure 2.** Output of a NLU elaboration with IBM Watson.

**Watson (IBM):** Watson is a NLU framework which allows building chatbots using a web-based user interface or using the most common programming languages, providing different SDKs. The users can create custom models for specific domains, only providing a relatively small set of examples to train the NLU engine. Once intentions or entities are recognized, the context allows storing data and reuse them in following dialogue interactions. Using the intuitive tree structure, it is possible to build deep and articulate dialogue frames. Once intentions and domains are identified, they are stored in a *context* and they can be retrieved and manipulated in the following interactions with the user. Watson permits defining different skills and each of them recognizes just a group of specific intentions and entities; *skills* can be connected and disconnected, aiming to add or remove the virtual assistant capabilities in different domains. Another characteristic is that Watson can be easily integrated with a variety of communications services, and it offers support also for speech translation.

**Dialogflow (Google):** Previously known as Api.ai, it has recently changed its name to Dialogflow, and it is a platform which helps to develop virtual textual assistants and

quickly add speech capability. Like Watson, this service offers a complete web-based user interface for creating chatbots and, for complex projects, it is possible to use a vast number of APIs via Rest or SDKs for several programming languages. Several intentions and entities can be created by providing examples. Sophisticated dialogue between the virtual assistant and the user are allowed by the particular focus that Dialogflow gives to the entire context, storing important information that can be found not only in the last interaction but in the entire course of the dialogue.

**Luis (Microsoft):** Similar to the previous platforms, Luis is a cloud-based service available through a web user interface or API requests. While allowing the users to develop their own models, Luis offers many pre-trained domain models that include intentions, examples and several entities. While the capability of Luis are similar to the ones belonging to the other services, the management of the dialogue flow results is more intricate and difficult to handle. Massive test support, sentiment analysis, speech recognition and other tools, which facilitate the development of a virtual assistant, are supported.

**Wit.ai (Facebook):** Wit.ai (bought by Facebook in 2016 and freely available) is quite different compared to other platforms. In fact, there is no concept of intent, but every model in Wit.ai is an entity and there are three different types of "Lookup Strategy" for distinguishing between different kinds of entities: *"Trait"*, when the entity value is not inferred from a keyword or a specific phrase in the sentence (in a similar way with respect to the intentions in the other platforms); *"Free Text"*, when there is the need to extract a substring of the message and this substring does not belong to a predefined list of possible values; *"Keyword"* is used when the entity value belongs to a predefined list, and we need a substring matching to look it up in the sentence. Moreover, while Wit.ai has a web-based user interface and API support like other platforms, the use of SDKs is quite limited, given that they cover only a features subset of the API.

## 4. Case Study

The Italian company *Mega Italia Media*, acting in the e-learning sector, is the owner of *DynDevice*, a cloud-based platform which provides online courses about "workplace safety" for thousands of users. While a messaging system which allows the users to contact human operators is already present on their platform, the recent increase in the demand of online courses lead them to decide to integrate their service with a chatbot which can answer the most frequently asked questions. Aiming at developing a chatbot that satisfies the company requirements, we have to find the best cloud-based NLU service on the market. Therefore, we analyzed the performance of different platforms exploiting real user requests (made in the Italian language), provided by the company.

Below, we show two examples of requests and their translation expressing the same intention. The first one is simple and the second one is more complex in order to show the variety and the complexity of the sentences coming from a real case of study:

- *"Corso scaduto è possibile riattivarlo?"*
  Course expired; is it possible to reactivate it?
- *"Buongiorno ho inviato stamattina un messaggio perchè non riesco ad accedere al corso "Lavoratori - Formazione specifica Basso rischio Uffici" probabilmente perchè è scaduto e purtroppo non me ne sono reso conto. Cosa posso fare? Grazie"*
  Good morning, I sent a message this morning because I cannot join "Workers - Specific training course about Low risk in Offices", probably because it is expired and unfortunately I did not realise it. What should I do? Thank you

In general, the requests made by the users are quite short, in fact 75% of them contains less than 50 characters. However, there are a few outliers with more than 500 characters (such as the example above) or less than 10.

### 4.1. Data Collection

Our dataset is composed by conversations between users and human operators occurred from the start of 2018 to the end of 2019. We decided to use only data from these two

years because, as the e-learning platform is constantly evolving, the requests from the users also keep evolving. For example, when a new feature is added to the platform, students might ask some questions about it and this generates a new class of requests. On the other hand, the resolution of bugs or the renewal of features can cause the disuse of one or more classes of requests.

*4.2. Data Classification*

As explained in Section 3, training NLU services on custom tasks requires a training dataset of labeled examples. The creation of this dataset, given that requests were not classified, was done manually. In the classification phase, we randomly extracted 1000 requests and showed them one by one to an operator who performed the manual classification using a command line interface. A tuple made by the request and the label is then saved in the dataset. All the sentences extracted from the conversations were anonymized. This process was also made by the operators who performed the classification, in fact when the text shown to the operator contains sensitive data, such as names, addresses, telephone numbers, etc., the operator has to replace this information with aliases which cannot lead to the identification of the user. To replicate a real interaction with a user, no other pre-processing operations (such as removing typos or special characters) were performed.

During the creation of the dataset, some requests were rejected. The most common cases are requests that were references to emails or phone calls that occurred previously between users and operators; meaningless requests, when the users typed in some text in the chat just to try it. In total, 33.1% of requests were excluded.

As we show in Figure 3, the examples belonging to the subgroup of the principal intentions are 551. One-hundred eighteen requests were too specific and could not be linked to an intention but, instead, they could only be directly satisfied by the tutors. A small set of requests (37) consists of text with a higher number of characters than the sentence length constraint imposed by the platforms, and that is why these examples were removed from the dataset. Therefore, the actual number of examples that can be used to build the training set and the test set is 514. The selected intentions are: I1, Human Operator; I2, Greetings; I3, Stopping the course; I4, Generic Issue; I5, Everything is ok; I6, Correction; I7, Number of attempts; I8, Certificates; I9, Deadlines. While there are common intentions such as initial and final greetings or simply the confirmation that the issues presented have been solved, some of them are quite specific for our application, such as requests of stopping an e-learning course or asking about the release of the certificate, the correction date, etc.

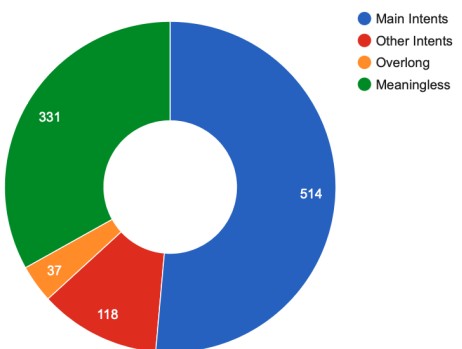

**Figure 3.** Requests made to the e-learning platform between 1 January 2018 and 31 December 2019.

Figure 4 shows the distribution of the 514 sentences classified over the intentions selected, which ranges from a minimum of 22 up to 190.

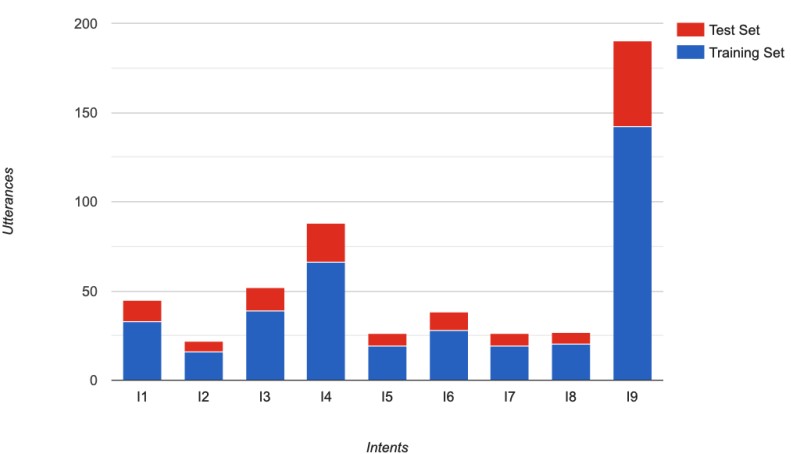

**Figure 4.** Number of intentions that compose the training and the test sets in Experiment (A).

## 5. Evaluation Experiments

As already mentioned in Section 3, we selected Watson, Dialogflow, Luis and Wit.ai, which can handle Italian. These platforms were evaluated in the free version because, while there is a limit in the number of APIs calls that can be made (in a day or in a month, depending on the service), there are no limitations in the services provided. We evaluated the ability to recognize the underlying intent of every message sent by users as described in Section 4. To evaluate the results as thoroughly as possible, we designed four different experiments. Experiment (A) aims at evaluating the performance on the whole training set; in other words, we used all the examples available to train the NLU platform. In Experiment (B), we studied the performance of the different systems at the increase of the number of examples provided to train the platforms; the test set is fixed for all the trials. Experiment (C) tests the response time of each platform. Finally, in Experiment (D) we tested the robustness of the services analyzed, so we built different test sets with an increasing number of misspellings in each example, and then we calculated the error rate of every single test set.

### 5.1. Training and Test Sets

To carry out the first experiment (A), we used data divided by main intentions. We divided our dataset in two: a training set, made by the 75% of the whole collection of examples, and a test set made of the remaining 25% of the elements. Figure 4 presents an overview of the distribution of the intentions among the training and test sets.

In Experiment (B), we used the same test set for nine different trials, in which we build nine subsets of the training set previously defined for Experiment (A). The first training subset was created by extracting 10% of the elements, keeping the same distribution of the intentions of the entire training set. The second one consists of 20% of the examples, keeping all the instances included in the first one and adding another 10% of examples for each intention. This process was repeated progressively, increasing by 10% of examples, until we reached the 100% of the initial training set. The composition of training subsets and test set is shown in Table 1.

In Experiment (C), we used the dataset built for Experiment (A). While the training set remained unchanged, in order to evaluate the response time using a considerable amount of requests, we created an extended test set replicating our 132 instances several times in order to reach a total of 1000 test requests.

As for Experiment (D), we also used the dataset built for Experiment (A). We used the training set to train the services, and we selected 25 test examples whose intentions were recognized correctly by all the platforms. With those elements, we created 30 new test sets: in the first new test set, we changed a character randomly for each example belonging

to the test set. To build the second test set, we used the test set just to create and change another character randomly for each example belonging to the test set. In this way, we created 30 different test sets, where the last test set had, for each example, 30 characters changed compared to the initial test set.

**Table 1.** Number of elements for each training subset and test set used in Experiment (B).

| Dataset | 10 | 20 | 30 | 40 | 50 | 60 | 70 | 80 | 90 | 100 |
|---------|----|----|----|----|----|----|----|----|----|-----|
| Training | 39 | 76 | 116 | 153 | 192 | 228 | 265 | 305 | 342 | 382 |
| Test | 132 | 132 | 132 | 132 | 132 | 132 | 132 | 132 | 132 | 132 |

*5.2. Experimental Design*

The number of examples for some intentions is quite small, and it is not possible to build a k-fold cross-validation, which is the most stable way to evaluate the performance of a machine learning model. Thus, for Experiment (A) we randomly sampled the training set and the corresponding test set elements (as described in Section 5.1) ten times, obtaining ten slightly different configurations for training and evaluating our system in a more complete way with respect to using a single test set. In fact, we want to evaluate whether the differences among the NLU platforms in terms of performance are statistically significant. For this purpose, we used the Friedman test and then proposed the pairwise comparison using Post-hoc Conover Friedman test as shown in [49].

To analyze Experiment (B), we took the dataset with the F-score closest to the average of the F-score on all datasets and we created nine subsets, as reported in Section 5.1, Experiment (B).

For Experiment (C), we used a training set of Experiment (A) and we built a test set with 1000 elements by replicating our test instances (as we explained in Section 5.1). To analyze the response time of the different services we sent each example in the test set independently to the NLU engines and we collected the time lapses between sending the message and receiving the reply. We summarized the times measured in a report that contains average and standard deviation, and highest and lowest response time. We repeated this experiment three times at different hours of the day because the servers were located in different parts of the world and each server load may have depended on the time when we performed the test.

In Experiment (D), we examined all the test sets created and, for each one, we found out how many times the service recognizes the correct intent, and then we calculate the error rate on all test sets analyzed.

We developed a Python application which uses the SDKs provided by the owners of the platforms with the aim of training, testing and evaluating the performance of each service. Although Wit.ai supports Python programming language, its instructions set is quite limited. To solve this issue, we implemented a program for invoking a set of HTTP APIs. Our application receives two CSV files as input (the training set and the test set), and outputs a report. The application is made by three modules working independently:

- The Training module: for each intention considered, all examples related to it are sent to the NLU platform and then the module waits until the service ends the training phase.
- The Testing module for each element in the test set, the module sends a message containing the user's utterance and the application waits for the response of the NLU platform. All the platforms analyzed report the results of the intent identification process and the confidence related to the prediction, therefore this module can compare the prediction made by the system and the correct intention. We want to underline that our application sends and analyzes every element of the test set (i.e., every user request) independently. In this module, there is an option that allows testing the response time using a timer which is activated before sending the message, and it is ended when the service reply arrives.

- The Reporting module, which is responsible for reading the file produced by the previous module and for calculating the performance metrics. The metrics calculated are Recall, Accuracy, F-score, and Error rate. This module also allows saving a report containing the response time in order to compute the average, the standard deviation, and the maximum and minimum response times.

## 6. Results and Discussion

As mentioned in Section 5, we created four different experiments. In experiment (A), we studied the performance on the training set with all elements available, in Experiment (B) instead, we evaluated the results with the increase of training set instances, in Experiment (C) we measured the response time of all platforms selected and in Experiment (D) we evaluated the robustness of services.

### 6.1. Experiment (A)

The goal of the first experiment is to highlight the different performances of NLU platforms. For this reason, we expected performances which were not uniform among different NLU platforms but, on the other hand, we supposed that the outcomes would be similar on the different randomly built datasets described in Section 5.1 (from here called D1, D2 . . . D10).

Table 2 presents the performance for each service in terms of Error rate and F-score (We used the Python function f1_score belonging to the sklearn module. As the dataset was unbalanced, we set the parameter "average" equal to "weighted".) on the ten different datasets, while the last two columns show the average and standard deviation. The best result regarding F-score for each dataset is reported in bold.

**Table 2.** Results in terms of error rate (Err) and F-score (FS) for the ten datasets created for Experiment (A). In the last columns, we also report the average (AVG) and the standard deviation (StD) across all datasets.

| | | D1 | D2 | D3 | D4 | D5 | D6 | D7 | D8 | D9 | D10 | AVG | StD |
|---|---|---|---|---|---|---|---|---|---|---|---|---|---|
| Watson | Err | 0.136 | 0.144 | 0.091 | 0.121 | 0.114 | 0.174 | 0.151 | 0.1364 | 0.167 | 0.144 | 0.138 | 0.023 |
| | FS | 0.857 | 0.851 | **0.907** | 0.878 | **0.887** | 0.822 | **0.845** | 0.863 | 0.832 | 0.857 | 0.860 | 0.02 |
| Dailoflow | Err | 0.129 | 0.144 | 0.144 | 0.106 | 0.121 | 0.151 | 0.159 | 0.106 | 0.136 | 0.144 | 0.134 | 0.017 |
| | FS | **0.877** | **0.851** | 0.856 | **0.894** | 0.875 | **0.857** | 0.844 | **0.893** | **0.871** | **0.863** | **0.868** | **0.016** |
| Luis | Err | 0.174 | 0.197 | 0.144 | 0.167 | 0.159 | 0.174 | 0.182 | 0.167 | 0.144 | 0.182 | 0.169 | 0.016 |
| | FS | 0.825 | 0.785 | 0.853 | 0.823 | 0.840 | 0.828 | 0.815 | 0.831 | 0.851 | 0.819 | 0.827 | 0.018 |
| Wit | Err | 0.227 | 0.273 | 0.280 | 0.273 | 0.265 | 0.235 | 0.311 | 0.258 | 0.273 | 0.318 | 0.271 | 0.027 |
| | FS | 0.778 | 0.716 | 0.692 | 0.708 | 0.706 | 0.757 | 0.671 | 0.743 | 0.718 | 0.656 | 0.714 | 0.035 |

As we supposed, there are several differences among the selected platforms in terms of general performance. The average F-score is quite similar between Dialogflow and Watson, which is over 0.86; however, Watson has a greater standard deviation. Luis' results are slightly worse than the other two with a 0.82 F-score, while Wit.ai obtains the worst performance (slightly above 0.71). In our datasets, neither Luis nor Wit.ai managed to overcome Dialogflow or Watson in terms of F-Score. We can see that most platforms have low standard deviation (StD) and that their results are stable considering our ten different datasets.

The number of examples is not the same for all the intentions in our evaluation. Consequently, the performance of the NLU services can be quite different among the intentions, as we can see in Figure 5 where we present separately the F-score outcome for every single intention. The Intention ID and number of examples used to train the services are shown in the legend of the figure. Looking at these diagrams, we notice how intention I9, which uses the highest number of elements, has one of the highest median and lowest dispersion among all services. I2, on the other hand, is the intention trained with the lowest

number of examples and, for this reason, its performance considerably varies among the different NLU platforms. Nevertheless, in all of them the median is far below 0.8 and the boxes are very stretched.

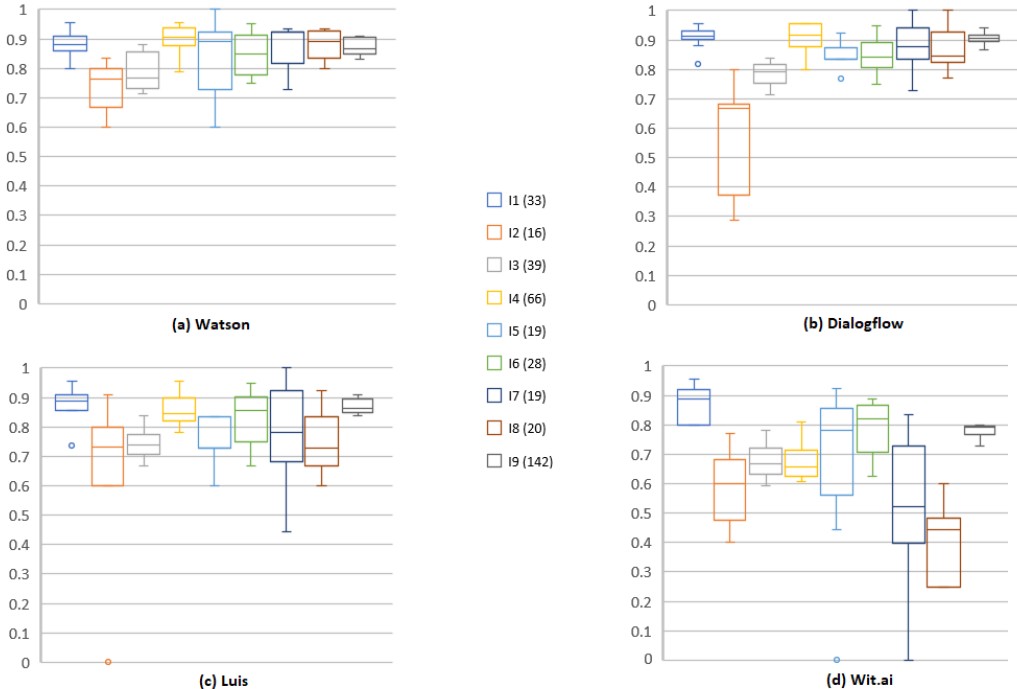

**Figure 5.** Results of Experiment (A) in terms of F-Score of each of the nine intentions for all the tested platforms (i.e., (**a**) Watson, (**b**) Dialogflow, (**c**) Luis, (**d**) Wit.ai).

Dialogflow has better overall results and the boxes are shorter than other platforms, but it presents an anomaly: the intention I2 has a significantly worse median value compared to Watson and Luis and its dispersion index is also very high. This may be due to the fact that Dialogflow needs a slightly higher number of training instances to achieve good performance, compared to the others. The outcomes of Watson and Luis are pretty good, with the median value always over 0.7. Wit.ai achieves the worst performance among the platforms tested. In fact, two of the intentions tested have a median value under 0.6 while only the other two intentions manage to reach a median value over 0.8. Furthermore, most of the intentions tested have a very high dispersion index.

We performed the Friedman test to understand if the differences in performance produced by NLU platforms selected are statistically significant. Figure 6 presents the pairwise comparison using the Post-hoc Conover Friedman test. We can observe that with $p < 0.001$, Watson and Dialogflow perform better than Luis and Wit.ai. These performance differences are significant according to Post-hoc analysis, while the difference between Dialogflow and Watson is not statistically significant using the same $p$-value ($p < 0.001$).

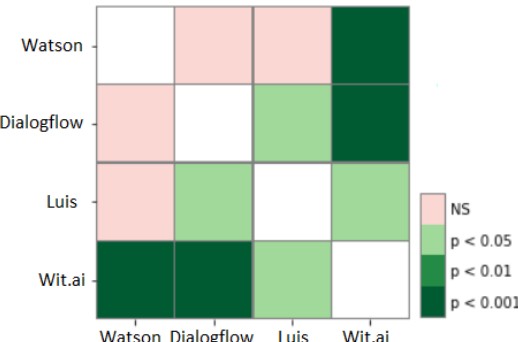

**Figure 6.** Results of the Post-hoc Conover Friedman test on the results of the tested platforms in Experiment (A). NS represents not significant.

### 6.2. Experiment (B)

In the second experiment, we selected the first dataset D1 and we split it into nine training subsets (as described in Section 5.1); we used the same test set to evaluate each of the nine models. We expected that increasing the size of the training set would correspond to an increase in the performance until they reached the same F-score obtained on the entire training set. Figure 7a confirms our assumption showing that, as the training percentage increases, so does the F-score of all four platforms. We can also see that the curves of Watson, Dialogflow, and Wit.ai grow quite quickly until we reach 40% of training instances and then fluctuate or grow slowly, whereas Luis rises steadily up until it reaches its maximum. Figure 7 shows the error rate for the different training set percentages. We can see that Watson and Dialogflow have significantly higher performance using fewer training instances (10% and 20% of the whole training set) compared to Luis and Wit.ai. Moreover, we can observe that, starting from the model trained with 50% of the training instances and going up, the difference in terms of error rate gradually decreases.

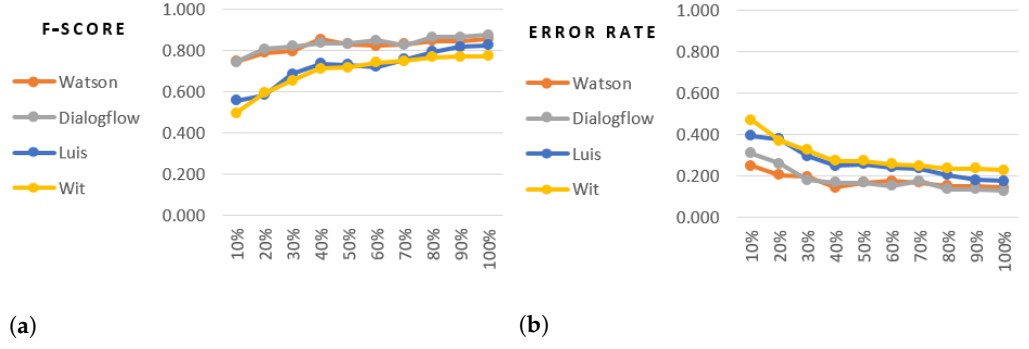

(**a**)           (**b**)

**Figure 7.** Results of Experiment (B) considering different percentages of the training set, in terms of F-Score (**a**) and Error Rate (**b**).

Finally, Figure 8 presents the number of the intentions which are correctly identified (ok), the ones which are incorrectly identified (ko) and those who were not found. In the calculation of the error rate, the "not found" intentions are considered alongside the wrong ones. An important difference among the platforms is that while Watson and Luis always provide a classification, Wit.ai and Dialogflow do not produce a specific prediction unless the confidence associated with the intention is over a certain threshold, therefore producing a higher number of "not found". The graph also shows a decrease of "not found" elements at the increase of the training set size.

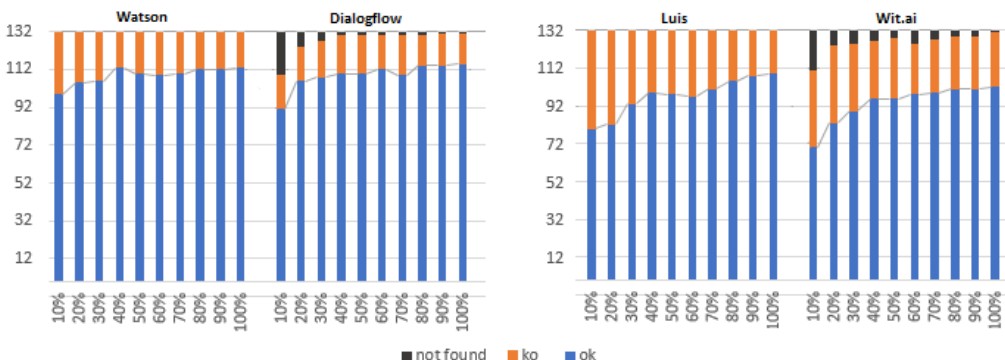

**Figure 8.** Number of correct identifications, incorrect identifications and identifications not provided in Experiment (B), considering different training set percentages.

### 6.3. Experiment (C)

In some specific applications such as Spoken Dialogue Systems (SDS), a short response time might improve the user experience and the general usability of the system.

In this experiment, we want to understand how much the response time varies among the various platforms. Since the servers that run these platforms are not in the same location, we executed the test three times in three different moments of the day. In Figure 9, each column is the average of the response times on the entire test set, while the yellow line represents the average of the three executions during the day. The results are expressed in milliseconds, and we use Rome Time Zone (CEST) to express times. Luis is the fastest platform, while Watson and Wit.ai are the slowest and Watson presents similar response times during the day. Wit.ai's performance seems to suffer from some excessive servers loading during particular times of the day; in fact, it has excellent results in the experiment performed at 11 p.m. while in the remaining two tests it shows the slowest response times. It should be noted that all platforms have average response times for each moment of the day under 400 ms, which is a reasonable response time for a messaging system.

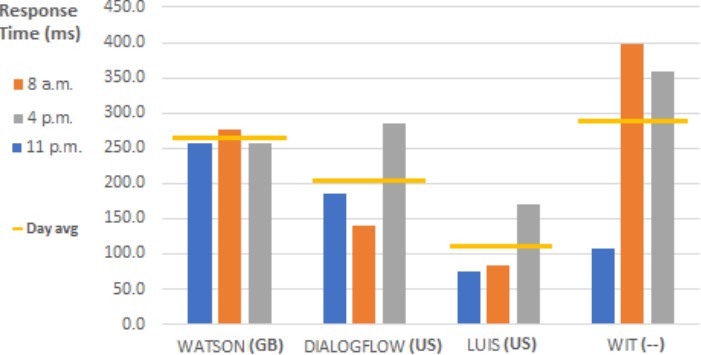

**Figure 9.** Results of Experiment (C) in terms of response time. The server location is reported next to the platform name.

### 6.4. Experiment (D)

As described in Section 5.2, here we want to evaluate the spelling errors robustness of NLU services; in order to do so, we created 30 datasets with an increasing number of random spelling errors for each of the examples contained. As we explained in Section 4, given that 75% of the users' messages contains 50 characters or fewer, we measure the robustness of our services even with a very high percentage of errors.

As shown in Figure 10, Watson is the most robust platform, and it maintains an error rate below 0.1 with fewer than 10 misspellings and an error rate under 0.4 with 30 spelling errors. The other three platforms have similar trends. Luis maintains an almost constant

error rate close to 0.5 when the spelling errors are between 20 and 30. Wit.ai and Dialogflow appear to be the least robust platforms in this experiment, in fact, their error rate exceeds 0.6.

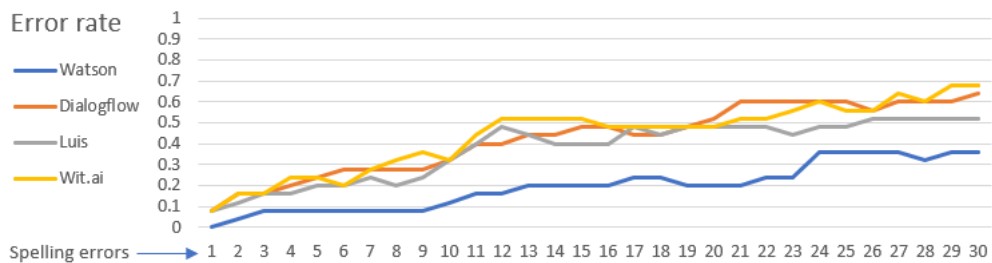

**Figure 10.** Results of Experiment (D) in terms of Error Rate at the increase of the spelling errors.

*6.5. Overall Evaluation*

In Table 3, we report an overall comparison of the results obtained by the different platforms in the Experiments (A), (B), (C) and (D).

**Table 3.** Overall comparison of the platforms tested considering the Experiments (A), (B), (C) and (D).

| Experiment | Requirement | Watson | Dialogflow | Luis | Wit |
|:---:|:---|:---:|:---:|:---:|:---:|
| (A) | F-Score greater than 0.8 | ✓ | ✓ | ✓ | × |
| (B) | Good performance with few training instances | ✓ | ✓ | × | × |
| (C) | Response times always under 0.5 s | ✓ | ✓ | ✓ | ✓ |
| (D) | Spelling errors robustness | ✓ | × | × | × |

In Experiment (A), all platforms, except Wit.ai, obtain good results, with a F-Score greater than 0.8. However, considering Experiment (B) Dialogflow and Watson have remarkably better results when the training set is very small. In terms of response time (Experiment (C)), the difference between the platforms selected is not particularly important. In fact, all platforms have a response time lower than 500 ms and a difference of 50 ms (such as the one between Watson and Wit) is not perceptible by the users. Considering Experiment (D), we find that Watson has a much greater robustness as concerns the spelling errors, while the other platforms have, on average, an error rate which is almost twice as high as the one obtained by Watson. Therefore, the extensive research conducted in the previous sections pushed us to choose IBM Watson as the NLU engine for this project since it is very robust, and it has outcomes comparable to Dialogflow.

## 7. System Implemented and Real-World Evaluation

*7.1. System Implemented*

DynDevice is the e-learning platform of Mega Italia Media; it is entirely written in PHP and students can use the chat system to send requests to the tutors. The chatbot system is designed to minimize the changes that have to be made to the existing system. It needs to be able to be easily deactivated in order to redirect the conversation to the human tutor if necessary.

Before the introduction of the virtual assistant, the student had to write the request in the chat interface, then the request was sent to the human tutor who answered through the same chat interface. This flow is shown in Figure 11 highlighted in the red dots box. The new system follows a new execution flow, highlighted in Figure 11 with blue dashes, and each step is enumerated:

1.   The chat is opened and the system sends a signal to the conversation manager.

2. The conversation manager creates a thread representing an instance of a conversation between a student and the NLU engine. Therefore, for each student who opens the chat, the conversation manager creates a dedicated conversation instance; this allows the system to manage multiple conversations simultaneously. Each instance is independent from the others.

3. The student enters the text into the chat interface. The request is sent to the conversation instance.

4. The conversation instance receives the student's request and checks the integrity of the sentence.

5. The conversation instance sends the student's request to the NLU engine using a specific API.

6. The NLU engine returns a message containing one or more intentions recognized with their confidence scores to the conversation instance. The NLU engine often predicts more than one intent, but the confidence allows distinguishing which intention is more likely than another.

7. The conversation instance scans all the intentions recognized and selects the one which has the highest confidence and builds the corresponding answer. If the student's request was unclear for the NLU engine, it is possible to have more intentions with similar confidence values, in this case, a disambiguation message is generated. A disambiguation message is a special request that the conversation instance sends to the student asking to specify, from a list of possible intentions previously selected, the correct one. Whether the intention of the student's request was evident, or the student responds to a disambiguation message, the conversation instance selects the correct answer and sends it back to the student.

8. The student receives the answer and (s)he can enter a new request into the chat interface and send it to the conversation instance.

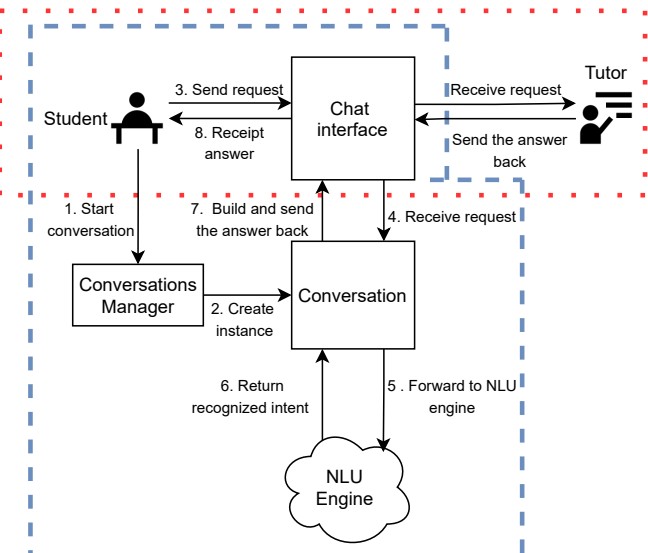

**Figure 11.** Architecture of the implemented system.

The chatbot may be unable to answer for various reasons; for example, the NLU engine cannot recognize the underlying intention of a request because it was not trained to recognize that specific intent, the student sent an unclear sentence, or the student asked to contact a human tutor. To guarantee the student's support, if the system implemented cannot provide help to the student, it deactivates the flow described, eliminates the conversation instance and redirects the student's conversation to the human tutor. When the dialogue is redirected to the human tutor, the system follows the simple execution flow shown in Figure 11 in the red dots box.

*7.2. Real-World Evaluation*

We monitored and collected the data of the system implemented, which is currently running in production, not only for *Mega Italia Media* but also for other 6 companies which offer e-learning courses and provide the same chatbot to their users.

From 1 November 2020 to 1 January 2022, more than 1400 messages were exchanged with users. Only 7% of them requested to talk with a human operator after interacting with our chatbot. In Section 4.2, we specified that the sub-group of the intentions selected covered 82.4% of the students' requests in the dataset extracted. With the new data coming from the performance evaluation, we have calculated that more than 93% of the requests belong to the 9 main intentions we selected for our experiments.

In Figure 12, we show the distribution of the main intentions recognized by our system during the evaluation period. To evaluate the system performance, the possible cases (which are also shown in Table 4) are the following:

1.  **Success**: the NLU engine recognizes the underlying intention and the chatbot provides the correct answer to the student. In Table 4, we do not classify the Human Operator intent recognition as Success. In fact, this intention can only be triggered after the chatbot fails to recognize a user request.
2.  **Disambiguation necessary**: the NLU engine does not provide a unique intention or the request is unclear, so the chatbot builds a disambiguation message as described previously. The student selects the correct intention among a subgroup of probable intentions. Therefore, the chatbot determines the correct answer and sends it to the student.
3.  **Fail and redirect to human tutor**: the NLU engine recognizes the wrong intention and the chatbot consequently sends an incorrect answer, therefore users ask to talk to a human operator; in this case, the chatbot does not attempt to satisfy any request and the user is immediately redirected to a human tutor.

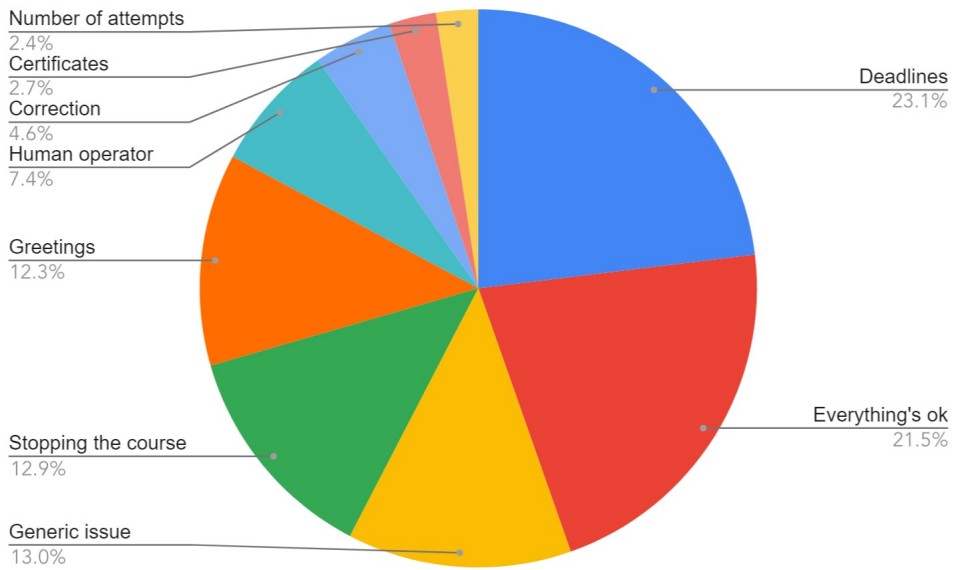

**Figure 12.** Pie chart of the main intentions recognized by our chatbot during the performance evaluation (from 1 November 2020 to 1 January 2022).

**Table 4.** Results of the intention recognition during the performance evaluation from 1 November 2020 to 1 January 2022.

| Occurrence | Percentage |
| --- | --- |
| Success (Without Human Operator intention) | 78.9 |
| Disambiguation | 15.1 |
| Fail and Redirect to Human Tutor | 6.0 |

Nevertheless, we cannot directly compare these results with the ones obtained in the platforms comparison. In fact, while in the studies conducted in the previous sections the intention with the highest confidence is the one considered predicted, in the real-world implementation, the NLU engine provides an ambiguous prediction of the intention, the system implemented asks the student to manually indicate the intention from a list. However, given that a vast majority of the requests are satisfied and 79.0% of the intentions are recognized, we think our results are stable and promising even in the real world environment and with a larger test set.

## 8. Conclusions

The e-learning sector has grown enormously during the COVID-19 pandemic, and therefore having artificial intelligence tools to support employees has become necessary. A virtual assistant that responds autonomously to a large group of users' requests can reduce the effort of human tutors and increase the scalability of an e-learning platform. This paper aims to describe the implementation of a chatbot that fits the company's needs. The first step to implement a chatbot consists of choosing the best platform on the market through an accurate and severe comparison. Thus, we presented four different experiments which compared, from different points of view, the ability of different cloud-based NLU platforms to recognize the underlying intent of sentences, referencing an Italian dataset from a real business class. After that, we integrated the chatbot in the assistant system, and we collected data concerning the production outcomes. Finally, we compared the performance of the system implemented with the results obtained by different NLU platforms.

In our analysis, Dialogflow shows the better overall results in experiment A; however, Watson achieves a similar performance and, in some cases, it outperforms Dialogflow. Luis also performs well, but in no case does it provide better results than the services already mentioned. In experiment A, Wit.ai had the worst results. Experiment (B) proves that both Dialogflow and Watson can achieve impressive results even when trained with just 40% of the whole training set. Experiment (C) shows that Luis is the fastest platform to recognize the intention associated with an input sentence. The response time of all platforms is on average below 400 ms, which can be considered acceptable for a chat messaging system. In Experiment (D), we notice that Watson is the most robust service when the sentences contain spelling errors. In fact, Watson's error rate is less than 0.3 when 30 characters are changed, while the other platforms in the same case have an error rate superior to 0.5.

In the light of the considerations above, in order to build a chatbot able to answer questions in Italian on an e-learning platform, the best NLU services are Watson and Dialogflow if we consider only raw performance in terms of F-score or Error rate. In the specific case of *Mega Italia Media*, the users of their e-learning platform are students with a wide variety of language skills. In this context, Watson is probably the best choice because it is the most robust service and the performance difference in terms of F-score is not statistically significant compared to Dialogflow, as shown in Section 6.1.

The system implemented achieves satisfactory results, with 79.0% of the users' requests satisfied during an evaluation which lasted more than a year. This paper shows that developing a virtual assistant quickly, with little data, achieving satisfactory performance and building a robust solution is possible using a cloud-based NLU platform. The significant reduction in the development times is mainly due to the drastic reduction in data

collection and preprocessing effort, which usually represent crucial and time-consuming tasks in this type of projects.

　　　Although we presented an application in the context provided by *Mega Italia Media*, similar behavior and performance are obtained in other contexts, such as the realization of a chatbot for a mobility transport domain and an Italian broadcasting company. These case studies are in the Italian language, so in the future we are planning to define an exhaustive linguistic analysis and compare the performance in other domains, with other languages. Finally, we want to use the results of the new analysis to improve our system features and increase their performance.

**Author Contributions:** Conceptualization and Supervision: L.S., A.E.G. and I.S.; Methodology: M.Z. and I.S.; Validation, L.P., I.S. and A.E.G.; Data Curation: L.S. and M.Z.; Software and Investigation: M.Z. and M.C.; Writing—Original Draft Preparation: M.Z., L.P. and M.C.; Writing—Review & Editing: L.P. and I.S. All authors have read and agreed to the published version of the manuscript.

**Funding:** This work was supported by MIUR "Fondo Dipartimenti di Eccellenza 2018–2022" of the DII Department at the University of Brescia.

**Institutional Review Board Statement:** Not applicable.

**Informed Consent Statement:** Not applicable.

**Data Availability Statement:** The data presented in this study are available on request from the corresponding author. The data are not publicly available due to privacy concerns.

**Conflicts of Interest:** The authors declare no conflict of interest.

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
