# Peer review of "A Performance Comparison of Different Cloud-Based Natural Language Understanding Services for an Italian e-Learning Platform†"

_futureinternet, doi:10.3390/fi14020062_

Round 1

Reviewer 1 Report

1. The authors present interesting work, however, the evaluation of systems was done with the Italian language only. Could other languages benefit from your research and your results?
2. The introduction section presents a very comprehensive covid situation overview in Italy, that is not related to the solving task.
3. The contribution is vague: the compared NLU platforms are popular on the market and therefore often tested for various languages with various datasets. I hardly recommend highlighting the novelty of your research in the Introduction or Related Work parts.
4. I am concerned about the reliability of your results: the comparison of different NLU platforms is done on one small dataset of the specific domain. Can your findings still be valid if having different domains, new datasets? Is it only the use-case or your finding can be generalized?
5. During the pre-processing step, a lot of cleaning is done (some texts are filtered out, some intents are eliminated) which means that you end up with a synthetic dataset that does not represent the real-world scenario anymore. Probably this is the reason why the results are so poor when you evaluate the system with the real dataset (in Table 5 the success is only 41%). What is the purpose of cleaning the training/testing datasets used in the comparative experiments of NLU platforms? If I understand the cleaning was done manually: are you planning to automatize this step in your system? If yes, then it makes sense.
6. Why have you oversimplified your intent detection task leaving only 9 intents? Are you planning to have a small number of intents in your e-learning system as well?
7. It is unclear to me how could you increase the number of testing instances from 132 to 1000 in the C experiment (lines 301-303)? Have you used duplicates?
8. What is the average length of your texts? If you randomly change 30 characters (lines 304-311) how difficult does the task become then? Are you trying to imitate typos? But they are typically made by selecting the wrong button next to the close one on the keyboard.
9. From the user perspective: does 41% of success (in Table 5) mean that the system is still acceptable for students as the e-learning platform?

Author Response

We would like to thank the reviewer for his or her insightful comments on the paper. In the current version, we have made several improvements.

First of all, as requested by #Reviewer1 (Point 1 and 3), we revised the paper carefully, and we tried to make our contribution more clear. Although #Reviewer1 is right saying that there are several works comparing NLU platforms such as (Thorat et al., 2020; Lokman et al., 2018; Canonico et al., 2018), we would like to point out that in this paper we present a more detailed analysis that takes also into account performance, number of training instances, response time and robustness to errors.  

Moreover, most of the previous comparisons are related to the English language; we think that proposing new results for a different language, such as Italian, could also be interesting for many researchers that have to work in a language other than English. Furthermore, the advantages and disadvantages of the main NLU platforms for the Italian language have been highlighted, and we speculate that similar conclusions can also be valid for other languages different from Italian and not as common as  English. (#Reviewer1 Point 1)

In any case, while we obtained similar results for other Italian datasets, as  future work, we think that it would be definitely interesting to perform a detailed comparison of the main NLU platforms using several languages; in the paper, this is reported in the Introduction and in the Conclusions sections. The differences between our approach and the state-of-the-art has been pointed out also in the Related work section.

As requested by #Reviewer1 (Point 2) and #Reviewer3 (Point 3 and 5), the Introduction section has been thoroughly revised. While we drastically diminished the space involving the dynamics of COVID-19 in Italy, which led the e-learning sector to a remarkable growth and aroused the necessity for a chatbot for Mega Italia Media; we improved the description of chatbots introducing their main components from an architectural point of view explaining how they can be divided into categories. We also removed the definition of e-learning, which is not particularly relevant to the work.

As correctly highlighted by #Reviewer1 (Point 4), we can’t generally say that IBM Watson is better in all contexts. However, we observed the same behaviour and similar results in the two applications that we developed in other contexts for the Italian language (a chatbot for a local transport company and one for a TV broadcasting company). We will conduct a more in-depth analysis, considering more languages and contexts, as a future work.

Apart from the anonymization process, in order to recreate a real-world environment, no pre-processing has been performed to the users’ requests. This is stated (in a clearer way with respect to the previous version) in Section 4.2 (#Reviewer1 Point 5).

We made a more clear presentation of how requests were selected and manually classified. In the production environment, as we explain in Section 7.2, our 9 main intents cover more than 93% of the total requests (#Reviewer1 Point 6).

We better described how the test set of Experiment C was created (Sections 5.1 and 5.2); moreover, in order to test the response time with a larger number of requests, we simply replicated the elements of our test set and reach a total of 1,000 instances  (#Reviewer1 Point 7).

As for the robustness to spelling errors, we have reported the average length of our texts in Sections 4 and 6.4. Given that most of our requests are short (often less than 50 characters), we highlight that we have included a considerable amount of errors (up to 30 characters) with respect to the total length  (#Reviewer1 Point 8).

Regarding the issues for the “real world evaluation” pointed out by #Reviewer1 (Point 9), we agree that the results were presented in an unclear way. Therefore, we entirely rewrote Section 7.2 presenting a new evaluation that considers the activity of the chatbot from November 1st, 2020 to January 1st, 2022, considering more than 1,400 messages exchanged with the users. In a simplified description, we can say that our system correctly recognizes 79% of requests.

Reviewer 2 Report

The paper describes a real case study in implementing a chatbot that improves the flexibility and scalability of the assistance system implemented, allowing to deal with peaks of requests with few tutors. 

During the Italian lockdown, the e-learning market has grown considerably during the pandemic, and having artificial intelligence tools to support employees have become necessary.

This paper shows that developing a virtual assistant quickly, with little data, achieving satisfactory performance, and building a robust solution is possible using a cloud-based NLU platform.

The paper is nice to read, the bibliography is up to date and the four experiments seem to be useful to understand the efficacy of each NLU platform. 

Author Response

We would like to thank the reviewer for his or her insightful comments on the paper.

Reviewer 3 Report

The present manuscript deals with a significant issue. The authors have done a great job managing to process a considerable amount of data and explore a suitable solution for implementing a chatbot in Italian. Still, I have a few suggestions for improvement:

1) The figures need to be clearly and meaningfully described - please improve the labels.

2) The tables need to be modified to make them more clearly. A table is a place to summarise and present results, and its design and labels need to match this.

3) The authors define e-learning but do not define chatbots. Why?

5) There is a lack of discussion regarding chatbots and their implementation. This lowers the level of the scientificity of the paper and, at the same time, opens the question of methodology. Is it adequate to test individual tools in this way? How would the results change with a different implementation, a different set of questions? The literature work needs to be strengthened in the introduction and discussion to clarify and justify the methodology. The literature is readily available.

6) I would appreciate some more apparent "call to action" - describing a straightforward recommendation or conclusion.

7) I am not a native speaker - I don't know all the grammatical issues. But still - the text needs major language correction. Please pay attention to it.

8) I think the text would benefit from more work with tables summarising the various experiments and results. It would make the text clearer and better structured. And above all - more readable.

Author Response

We would like to thank the reviewer for his or her insightful comments on the paper. In the current version, we have made several improvements.

First of all, we improved the data and results presentation, introducing new figures and tables and improving the labels (#Reviewer3 Point 1 and 2).

As requested by #Reviewer1 (Point 2) and #Reviewer3 (Point 3 and 5), the Introduction section has been thoroughly revised. While we drastically diminished the space involving the dynamics of COVID-19 in Italy, which led the e-learning sector to a remarkable growth and aroused the necessity for a chatbot for Mega Italia Media; we improved the description of chatbots introducing their main components from an architectural point of view explaining how they can be divided into categories. We also removed the definition of e-learning, which is not particularly relevant to the work.

As for the question about methodology asked by #Reviewer3 (Point 5), we can say that we are considering several important aspects analysed by the work of (Luo et al., 2022), which indicates usability, customization and task performance as important guidelines to evaluate a chatbot.

#Reviewer3 (Points 6 and 8) pointed out that our analysis lacks a straightforward conclusion and a summarisation. In order to solve this issue, we introduced a new section (6.5, Overall evaluation) that summarises the experiments A, B, C and D into a unique table and provides a simpler description of the process behind the choice of IBM Watson as the candidate system for Mega Italia Media.

We have carefully revised the English (#Reviewer3, Point 7); however, due to the limited time available, we have not been able to hire your editing services for a complete check of the English language. If requested, we can foresee it for the final version of the paper, if accepted.

Round 2

Reviewer 1 Report

Thank you for considering my remarks. 

Author Response

Thank you for all your advices and suggestions.

Kind regards,

The authors.

Reviewer 3 Report

I think that the work has improved substantially. Still, I would see it as applicable for the authors to devote space to defining "What is a chatbot". There are many definitions, and subscribing to a particular tradition or authors would be relevant. The current description tells what parts a chatbot consists of or what it is supposed to do, but not what it is. While the discussion could be more complex, the paper is relevant and ready for acceptance once the definition of a chatbot and the relevant literature on the topic is completed.

Author Response

Dear reviewer,

on page 2 (highlighted in blue) we have introduced a space for defining a chatbot, with several references to the literature.

This manuscript is a resubmission of an earlier submission. The following is a list of the peer review reports and author responses from that submission.